# Emotional Intelligence as a Predictor of Motivation, Anxiety and Leadership in Athletes

**DOI:** 10.3390/ijerph19127521

**Published:** 2022-06-20

**Authors:** Isabel Mercader Rubio, Nieves Gutiérrez Ángel, María Dolores Pérez Esteban, Nieves Fátima Oropesa Ruiz

**Affiliations:** 1Departamento de Psicología, Facultad de Ciencias de la Educación, Universidad de Almería, 04120 Almeria, Spain; foropesa@ual.es; 2Departamento de Educación, Facultad de Ciencias de la Educación, Universidad de Almería, 04120 Almeria, Spain; mpe242@ual.es

**Keywords:** emotional intelligence, motivation, anxiety

## Abstract

Nowadays, emotional intelligence is not only understood as the recognition of our own emotions but also the regulation of these emotions. In the field of sports, the concept of sports leadership is increasingly relevant, understood as a behavioral and cognitive process closely related to sports success, based on interpersonal relationships, trust, respect and the feeling of coherence. In this study, we intend to analyze the relationship between sports success and emotional intelligence to verify their relationship and the influence of other variables such as sports anxiety. As a sample, we took a total of 165 active sportsmen and women studying for both undergraduate and master’s degrees related to the sciences of physical activity and sport. The expected results aim to demonstrate the relationship between emotional intelligence, sports leadership and sports anxiety.

## 1. Introduction

One of the main objectives of sport psychology is to identify those psychological factors that are relevant to sports performance, as well as the possibilities for their development [1]. Thus, within sport psychology, we find emotional intelligence to be one of the most novel concepts and lines of research of the 21st century, which nowadays is the subject to extensive popularity and research [2,3,4,5] and understood as the emotional inter or intrapersonal response of an athlete [6].

Therefore, emotional intelligence is considered to be a very relevant psychological skill in the field of sports, influencing both emotional control by athletes [7], as well as decision making and performance [8].

Despite the existence of two major models to explain and understand the concept of emotional intelligence, mixed models and models of ability, in this study, we specifically focus on models of ability [6], which are characterized by different sections or branches. Each of these has a great number of skills, which correspond to: Emotional perception—this is the ability to both identify and recognize one’s own and others’ feelings. It implies, therefore, an interest in and knowledge of different signs expression, sensations and the sincerity of emotions. Emotional understanding—this is about knowing, classifying and examining emotions in a retrospective way, both one’s own and others’ emotions. Emotional regulation—this is about capturing, analyzing and reflecting on emotions, in order to make the most of them, both interpersonally and intrapersonally. In this study, the TMMS-24 [9], which corresponds to an assessment measure created in line with the theoretical model of emotional intelligence [6], is used as an instrument for measuring emotional intelligence. Such an instrument is a self-report measure that attempts to assess perceived emotional intelligence, (i.e., a person’s own knowledge of his or her own emotional abilities).

In the field of sports, there are several studies that have used athletes as a sample to determine the role of emotional intelligence in sports situations. Thus, the results show that those athletes who have higher levels of emotional intelligence are more effective in competitions [10], so we can establish that there is a direct and positive correlation between this type of intelligence and sports performance.

The same situation is found for the self-concept of athletes; we found a direct and positive relationship between self-concept and the levels of emotional intelligence [10]. As with motivation, studies showed that high levels of emotional intelligence have a continuous and existing correlation with self-concept [11,12,13,14].

However, on the opposite side, we find a series of studies that, in the case of anxiety, demonstrate that the correlation between anxiety and emotional intelligence is negative, i.e., that those athletes who have high levels of emotional intelligence control their anxiety better, and their levels of anxiety are much lower [10,15,16]. The same occurs in the case of stress, which also has a negative correlation [17]. Athletes are sometimes faced with a large number of stressors that can cause high levels of anxiety [18]. Therefore, working on emotional intelligence through the proposal and consideration of skill models corresponds to an appropriate action to help athletes cope with sports anxiety.

Finally, as for sports leadership, it is currently an essential topic in terms of sports effectiveness [19], and despite the broad terminological framework that revolves around sports leadership, all definitions coincide by pointing this out as a process in which the influence of other members of the group occurs in order to achieve a common goal [20]. In these contexts, we are committed to a concept of sports leadership that can come from both the coach, a captain, and an athlete (whether professional or not). In this sense, several studies showed that emotional intelligence has positive implications for the emotional well-being of people in leadership positions, which positively influences team cohesion, satisfaction, identification, confidence and motivational climate [15,16,17,18]. In this work, we opted for a conceptualization of leadership [21] to understand leadership from both a social and task-oriented perspective, which is composed of a total of five primary factors: empathy, social support, influence on decision making, sports values, and task orientation.

In view of the results, as a research question, we proposed an analysis of whether emotional intelligence is a predictor of other issues related to sport psychology, such as anxiety, leadership, or motivation. If we train the emotional intelligence of athletes, are we also training their leadership or motivation skills? Can athletes be better in their sporting activities if we train their emotional intelligence?

Therefore, the aim of this study is to analyze whether emotional intelligence and its dimensions (attention, clarity and emotional regulation) are predictor variables of anxiety, leadership and motivation in athletes.

## 2. Materials and Methods

### 2.1. Method

The method used was correlational, corresponding to an ex post facto design and retrospective and comparative in nature, since the dimensions of EQ are compared with other types of variables, in this case dependent variables, corresponding to leadership, motivation and anxiety.

### 2.2. Participants

The sample was composed of 165 students studying undergraduate and master’s degrees related to physical activity and sport sciences.

The age of the sample was 20.33 years, with a standard deviation SD = 3.44. As for sex, 70.9% (*n* = 117) were male and 27.9% (*n* = 46) were female.

The sample size was calculated using the Soper a priori sample size calculator for structural equation models [22]. Thus, based on six observable variables, one latent variable, with an anticipated effect size of 0.30, probability level of 0.05, and desired statistical power level of 0.95, the minimum recommended effect size was 200 cases, meaning the number of participants in our study was close enough to the suggested population size. The questionnaire was administered to the four undergraduate courses in physical activity and sport sciences and to students studying master’s degrees in teaching (specializing in physical education) and sports science research.

### 2.3. Instruments

The instruments used in this study are as follows:The TMMS-24 [23] corresponds to a kind of submodel of emotional intelligence created from the model of [6], which some authors call the Spanish version of the model of [6]. For these authors, emotional intelligence is composed of three dimensions, such as the perception, understanding and regulation of emotions through a Likert-type scale [24]. A Cronbach’s alpha = 0.84 was obtained in this study. It is composed of twenty-four items, eight of which refer to one’s own feelings, eight to clarity and eight to the regulation of feelings, resulting in a Likert-type scale of self-knowledge that assesses three dimensions of emotional intelligence: attention to feelings, emotional clarity and emotion repair.

In addition, it shows a high reliability (Cronbach’s alpha) for each dimension (perception, a = 0.90; clarity, a = 0.90; regulation, a = 0.86).

Adequate test–retest reliability: perception = 0.60; comprehension = 0.70 and regulation = 0.83 [25].

The adaptation of the Sport Leadership Behavior Inventory was developed by [26]. This scale measures aspects of leadership related to social support, empathy, sport values, and influence in decision making and task orientation through a Likert-type scale. A Cronbach’s alpha = 0.94 was obtained in this study, whereas a Cronbach’s alpha score of 0.84 was obtained in other studies [27].The sport motivational mediators scale [28] measures the beliefs and mediators that predict motivation through a Likert-type scale. A Cronbach’s alpha = 0.71 was obtained for this work. The reliability of the original instrument is 0.75 for the first factor; 0.76 for the second; and 0.71 for the third [29].The SCAT [30] analyzes responses to a series of statements about how one feels in a competitive situation through a Likert-type scale. A Cronbach’s alpha = 0.107 was obtained in this study, whereas in other studies, Cronbach’s alpha was = 0.91 [31].

### 2.4. Data Analysis

The data analyses used in this study were descriptive statistics (mean, standard deviation and bivariate correlations), reliability analyses and structural equation modeling (SEM) in order to contrast the relationships established in the hypothesized model. Specifically, we applied Joreskog’s technique for the analysis of covariance structure [32,33] to a multiple indicators multiple causes (MIMIC).

To accept or reject the proposed model, a set of suitable indices were taken into account [34]: TLI (Tucker–Lewis index), SRMR (standardized mean root square residual) and RMSEA (root mean square error of approximation).

Thus, the suitable indices are: TLI value above 0.95; SRMR values below 0.06; and RMSEA values below 0.08. These analyses were carried out using SPSS version 26 and R statistical analysis software.

## 3. Results

The relationships between each of the dimensions of emotional intelligence (AEM/CEM/ERM) and their relationship with leadership (LIM), anxiety (AM), and motivation (MM) were analyzed.

As can be seen in Table 1, the correlations between the study variables were positive, reflecting the reciprocity between the study variables.

### Structural Equation Mode

The hypothesized model of predictive relationships (Figure 1) shows that the fit indices were adequate: *p* < 0.001, IFI = 1.033, TLI = 1.074, CFI = 1.00, RMSEA = 0.00 (90% CI = 0.050–0.061), SRMR = 0.26.

The relationships established in the structural equation model are specified below:(a)Emotional intelligence and motivation were positively correlated (β = 0.30, *p* < 0.001). This explains that, in this case, emotional intelligence is a predictor of motivation; therefore, the presence of this variable explains the existence of the other variable.(b)Emotional intelligence and anxiety were positively related (β = 0.02, *p* < 0.001). These results show that emotional intelligence also predicts the occurrence of anxiety in an explanatory manner.(c)Emotional intelligence and leadership were not positively related. However, the results do not establish that emotional intelligence is a predictor variable of leadership; therefore, the presence of emotional intelligence does not correspond with the appearance of high levels of leadership.

## 4. Discussion and Conclusions

The aim of the present study was to analyze whether emotional intelligence and its dimensions (attention, clarity and emotional regulation) are predictors of anxiety in athletes, leadership in athletes and motivation in athletes.

The results of this study show a direct and positive relationship for some questions but a negative relationship for others. Thus, focusing on the results that show a direct and positive correlation, we found a relationship between emotional intelligence and motivation, as other studies already showed [12,13,14].

The same is true for anxiety. Despite this, the results of this study do not coincide with results from previous studies, where, unlike our study, the three factors of emotional intelligence were not predictors of emotional intelligence [11,27,28,29]. Therefore, in this study, we focused not only on the importance of emotional intelligence in sport, but also on the need to include training and stimulation programs for athletes. The ability models of emotional intelligence show that it can be learned, acquired and improved [6], so we consider this issue essential in working with athletes.

However, it should be noted that the results from this study do not show a positive relationship between the three factors of emotional intelligence and leadership, unlike the results of previous studies, in which a direct and positive relationship was found [15,16,17,18]. The results of this study can perhaps be explained by its small sample size or the fact that we focused on leadership in general and not on each of the five main factors: empathy, social support, influence of decision making, sport values, and task orientation [30].

In short, the results of this study show a strong relationship between emotional intelligence and anxiety and between emotional intelligence and motivation. Future lines of research will aim to determine whether there are differences according to the type of sport practiced, as well as expanding sample size, since this was one of the limitations of this study. In this way, future studies will attempt to analyze differences according to the degree of professionalism of each sport played by future participants.

## Figures and Tables

**Figure 1 ijerph-19-07521-f001:**
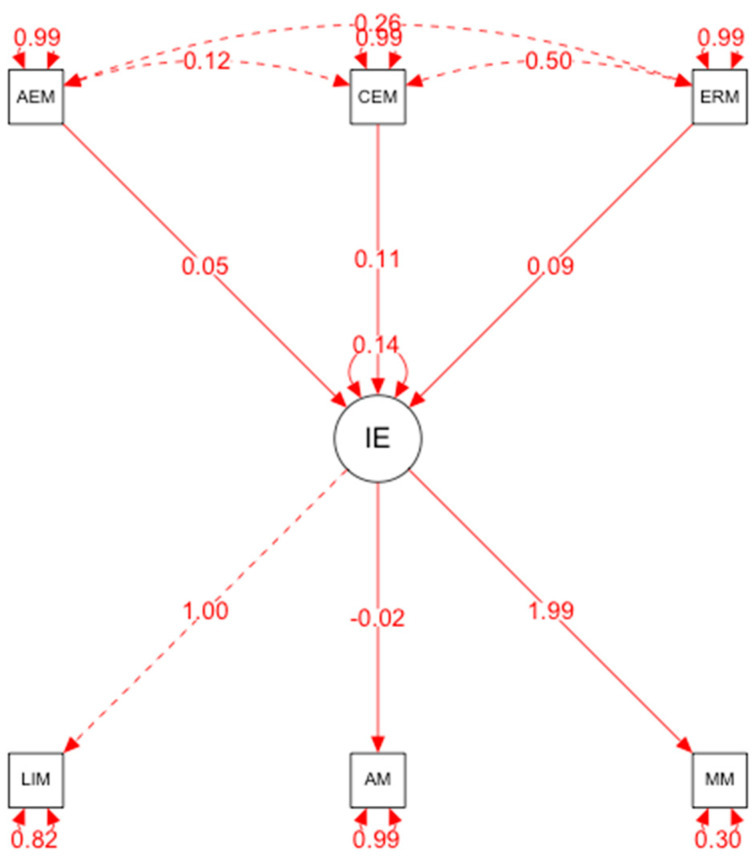
Structural equational modeling: AEM: emotional attention; CEM: emotional clarity; ERM: emotional regulation; IE: emotional intelligence; LIM: leadership; AM: anxiety; MM: motivation.

**Table 1 ijerph-19-07521-t001:** Preliminary analysis.

	1	2	3	4	5	6
1. Emotional attention (AE)						
2. Emotional clarity (CE)		0.127	0.263 **	0.150	0.321 **	0.181 *
3. Emotional regulation (ERM)			0.508 **	0.156 *	−0.069	0.321 **
4. Leadership (LIM)				0.157 *	−0.052	0.326 **
5. Anxiety (AM)					0.049	0.364 **
6. Motivation (MM)						−0.018

Note. * *p* < 0.05; ** *p* < 0.01.

## Data Availability

Not applicable.

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
