# Peer review of "Emotional Intelligence as a Predictor of Motivation, Anxiety and Leadership in Athletes"

_ijerph, 2022, doi:10.3390/ijerph19127521_

Round 1
Reviewer 1 Report
The manuscript entitled “Emotional intelligence as a predictor of motivation, anxiety and leadership in athletes” presents the study on the associations between emotional intelligence and motivation, leadership and anxiety in sport. Unfortunately, I have several major doubts regarding the manuscript.
#1. The better introduction of emotional intelligence model is needed. The model used by the Authors is based on trait meta-mood. It should be clearly stated that this is a dispositional (not ability) model of emotional intelligence. It is also hard to use EQ according to this approach to EI.
#2. Better description of leadership and anxiety in sport should be included. It will also help to better develop and justify the hypotheses on the role of EI in sports. It should be also justified why these variables were taken for the study.
#3. The Table 1 is not possible to understand due to lack of variables names. Similarly, coefficients symbols are lacking in p. 4, l. 119-122.
#4. The description of Figure 1 is needed. Are coefficients in Figure raw or standardized? Which is the reason of SC latent variable constructing?
#5. Discussion is very limited. Which are the possible explanations of the obtained results? Both those congruent with previous research and those which were contrary.
#6. Please clarify how the sample size was determined.
Author Response
Dear reviewer,
Thank you very much for your kind words and for your help, which has greatly improved our manuscript.
- Introduction: The content of the introduction has been reorganized, clarifying the conceptualization of emotional intelligence, the existence of two models, and the fact that we focus in this paper on a skill model and its characteristics.
In addition, a redefinition of anxiety (59-66) and leadership, including recent citations (75-79), has been included in the introduction. We have also included research questions and a rationale for why we measured these variables.
Research questions have been included and a rationale for why we measure these variables has been provided (80-84).
- Results: Table 1 has been modified, indicating variables and acronyms. The corresponding information has been incorporated in the x and y axes. And the coefficients have been added. Figure 1 has been reconfigured. The coefficients are standardized, and the SC variable is renamed IE (emotional intelligence) to make it easier to understand.
- Discussion: The results obtained and their possible causes have been explained. Limitations and future lines of research have been added. In addition, the sample size has been specified.
Reviewer 2 Report
The introduction needs a lot more thought and effort to bring it up to a good standard. The way it reads now is like a set of notes rather than a coherent series of paragraphs leading to the aims of the study and any hypotheses.
The method was adequate but some more details on the reliabilities and validities of each measure need to be included.
In the results Table 1 needs the axes to labelled properly so that the reader can see which variables are correlated with which other variables. That is the names of each variable should be on the x and y axes. The description of the results requires a bit more detail of what was found without interpretation.
The discussion is little more than a restatement of the results and does not integrate the findings into previous literature. This is the poorest section of the paper and needs the most work. The other sections introduction, methods and results requires some strengthening and detailing and editing.
Author Response
Dear reviewer,
Thank you very much for your kind words and for your help, which has greatly improved our manuscript.
- Introduction: The content of the introduction has been reorganized. We have included research questions and justified why we measure these variables (80-84).
- Method: The reliability and validity of the TMMS-24 (224-227) was added. The leadership measurement instrument: The adaptation of the Sport Leaderchip Behavior Inventory (231-232). The motivation questionnaire: The sport motivational mediators scale: SCAT (235-236). And the anxiety questionnaire (239).
- Results: Table 1 has been modified, indicating variables and acronyms. The corresponding information has been incorporated in the x and y axes. A brief explanation of the results has been added (282-289).
- Discussion: The results obtained and their relationship with previous studies on the subject have been explained (271-274-283-285).
Round 2
Reviewer 1 Report
The Authors corrected their manuscript according to the Reviewers' suggestion. However, I have some other suggestions:
#1. The model of EI which was used is Trait Meta-Mood model. According to references the Authors used Spanish version of TMMS, which measures: Attention to emotion, Clarity of Emotion and repair. This information is lacking in the manuscript. Moreover, the acronyms used in the figure are not described. Please, correct a description of TMMS and indicate clearly in the introduction that you have used this approach to EI (it is not ability approach - the measurement is via questionnaite, not using a test of EI).
#2. The Authors should describe which coefficients are presented in the Figure (unstandardized?)
#3. In the description of the results from SEM some symbols are missing (p. 5; l. 162-169; _).
#4. The sample size consideration should be connected with a priori power analysis, not the problem with administration of the study. The problem is whether the Authors gather a sample power enough to detect expected associations between variables. Please clarify.
Author Response
- TMMS, which measures: attention to emotion, clarity of emotion and repair. This information is missing from the manuscript.
- A description of TMMS 24 has been included in the introduction and in the instruments (43-53) (234-236)
- In addition, the acronyms used in the figure are not described.
- A legend has been included below the figure indicating the acronym used in the model
- Correct a description of TMMS and clearly indicate in the introduction that you have used this approach for EI (it is not a capability approach: the measurement is done through a questionnaire, an EI test is not used).
- Changed
- Authors should describe which coefficients are presented in the figure (not standardized?)
- The standardized coefficients are shown in Figure 1.
- Some symbols are missing from the description of SEM results (p. 5; l. 162-169; _).
- The missing symbols in the description of the SEM results are added (Lines 163 and 167 of p.5).
- Consideration of sample size should be related to an a priori power analysis, not to the problem of study administration. The problem is whether the Authors collect a sample powerful enough to detect the expected associations between the variables. Please clarify.
- It is clarified that the sample size has been calculated according to the a priori statistical power analysis (lines 95-103).
Reviewer 2 Report
The new sections that have been added contain some errors of English expression and grammar that require careful editing.
Author Response
Both the expressions and the translation have been revised.